# Zinc Recovery through Electrolytic Refinement Using Insoluble Ir + Sn + Ta + PdOx/Ti Cathode to Reduce Electrical Energy Use

**DOI:** 10.3390/ma12172779

**Published:** 2019-08-29

**Authors:** Ji-Hyun Kim, Jung Eun Park, Eun Sil Lee

**Affiliations:** Center for Plant Engineering, Institute for Advanced Engineering, Yongin-si 17180, Korea

**Keywords:** zinc recovery, insoluble catalytic electrode, electrical energy, recovery time, current density, activation polarization, current resistance

## Abstract

In this study, an alumina (Al) anode, a lead cathode, and insoluble catalytic cathodes (IrOx, PdOx, TaOx, and SnOx) were used as electrodes to enhance zinc recovery. The traditionally used iron electrode and insoluble catalytic electrodes were also used to compare the recovery yield when different types of electrodes were subjected to the same amount of energy. The lead electrode showed over 5000 Ω higher electrode resistance than did the insoluble catalytic electrode, leading to overpotential requiring higher electrical energy. As electrical energy used by the lead and the insoluble catalytic electrodes were 2498.97 and 2262.37 kwh/ton-Zn, respectively, electrical energy can be reduced by 10% when using an insoluble catalytic electrode compared to that when using a lead electrode. Using recovery time (1–4 h) and current density (100–500 A/m^2^) as variables, the activation, concentration polarization, and electrode resistance were measured for each condition to find the optimum condition for zinc recovery. A recovery yield of about 77% was obtained for up to 3 h of zinc recovery time at a current density of 200 A/m^2^, which is lower than that (about 80%) obtained at 300 A/m^2^. After 3 h of recovery time, electrode resistance (Zn concentration reduction, hydrogen generation on electrode surface) and overpotential increase with time decreased at a current density of 200 A/m^2^, leading to a significant increase in zinc recovery yield (95%).

## 1. Introduction

The zinc smelting process, urban mining, and plating industries produce zinc solutions containing various impurities (F, Cl, Mn, As, Mg, and Ca); however, methods to recover high-purity zinc from these solutions are actively being researched [1]. The most commonly used method involves solvent extraction, through which zinc solution and impurities are separated to produce a zinc electrolyte with high purity and high concentration. Afterward, the electrolyte is subjected to electrolyte refinement to recover high-purity zinc metal [2,3,4,5,6]. During electrolyte refinement, the electricity cost accounts for 50–80% of the production cost and is the main factor that reduces the economic feasibility of the zinc recovery process. High electric power is required for the use of lead (Pb) plates as electrodes during zinc recovery. Limitations of lead electrodes require a high overpotential, which contributes to the high electric cost and produces precipitates due to the low erosion resistance, thus reducing the electrode lifespan [7].

Titanium (Ti) electrodes with high erosion resistance and conductivity are, therefore, being used instead of lead electrodes; in particular, electrodes made of titanium plate with various insoluble catalyst coatings are being developed [8]. Insoluble catalysts include IrOx, PdOx, TaOx, and SnOx, where IrOx and PdOx are used as activators to improve the current efficiency, while TaOx and SnOx are used as a stabilizer and dispersant, respectively, to increase the electrode’s lifespan [9,10,11,12,13]. For example, in a IrOx/Ti electrode, Mn in the electrolyte is electrodeposited in an oxide form at the cathode surface to increase cell resistance, while F ions corrode the coated catalyst at the cathode surface, critically reducing the electrode efficiency and lifespan [14]. To improve the electrodes, TaOx–SnOx–PdOx oxides can be added as the main catalysts for the IrOx electrode to increase the electrode’s lifespan and current efficiency compared to an IrOx/Ti electrode [14].

Design factors for the zinc recovery process include the zinc purity, cathode cell voltage, electrical energy, pH, current density, zinc concentration, recovery temperature, and electrode spacing. Regarding the pH factor, a zinc solution was prepared using zinc oxide and sulfuric acid, and the zinc concentration was reported to be different depending on the pH. The reaction scheme of the reaction is as follows (Equation (1)):ZnO + H_2_SO_4_ → ZnSO_4_ + H_2_O.(1)

Pearson et al. (1981) showed zinc leaching concentrations according to the pH in industrial dust containing zinc by using the EAFD method. When the pH was 2, zinc was recovered to 85%–90%, and the zinc recovery was reduced by 80% as the pH was increased to 3–4 [15]. By lowering the pH, a high concentration of zinc solution could be produced. In addition, the current density, zinc concentration, recovery temperature, electrode spacing, etc. were known to promote zinc recovery by improving Zn ion transfer [15].

In this study, therefore, an alumina (Al) anode, a lead cathode, and insoluble catalytic cathodes, which can increase electrode stability and conductivity, were used to develop electrodes. By comparing the electrodes with the lead electrode conventionally used in zinc recovery and insoluble catalytic electrodes, the increase in recovery yield was identified for the given energy. Furthermore, the recovery time and current density of insoluble catalytic electrodes were used as variables to measure activation, concentration polarization, and electrode resistance in order to find the optimum condition for zinc recovery. 

## 2. Materials and Experimental Methods

### 2.1. Materials

For the lead electrode, lead plate (thickness: 1.0 mm, manufacturer: Metal Power) was purchased and cut into an electrode surface area of 100 mm^2^. The three- and four-component electrodes were purchased from Wesco Electrode. The Ti plate (thickness: 2.0 mm) was used and coated with three components (Ir + Sn + TaOx/Ti, IrOx:SnOx:TaOx = 45:45:10, wt%) and four components (Ir + Sn + Ta + PdOx/Ti, PdOx = 8–12, wt%) with a 20 µm thickness [14]. Electrolyte containing zinc ions was produced using sulfuric acid (H_2_SO_4_, manufacturer: Samchun, Seoul, republic of Korea purity: 95%) and zinc oxide (ZnO, manufacturer: Sigma-Aldrich, St. Louis, MO, USA purity: 99.9%). The composition of the solvent extract solution was (Zn—116.73 g/L, Na—0.43 g/L, Mn—0.41 g/L, and S—75.59 g/L).

### 2.2. Measurement Methods

The content of the coating material of the insoluble catalyst electrode was analyzed using scanning electron microscopy with energy dispersive X-ray spectroscopy (SEM/EDX) (model: Inspect F50, manufacturer: ThermoFisher, Hillsboro, OR, USA). Zinc recovery was measured with the Galvanic measurement method, using a multi-channel electrochemical measurement device (model: ZIVE BP2, manufacturer: WonATech, Seoul, Korea) by applying a certain amount of current to recover zinc in the anode. The Galvanic measurement method was used in the condition where the Al electrode was the anode and lead or an insoluble catalytic electrode was the cathode with the same electrode surface area (100 mm^2^). In this study, the conditions used were electric currents of various intensities (0.1–0.5 A), recovery time from 1 to 4 h, sulfuric acid concentrations of 3 M, zinc concentrations of 100 g-Zn/L-3 M H_2_SO_4_, electrode distance of 3 cm, electrolyte of 2 L, solution temperature of 40 °C, and stirring speed of 150 rpm. Polarization and resistance measurements were conducted using three-electrode systems with Ag/AgCl as the standard electrode, where a Pt electrode was the anode and a lead or an insoluble catalytic electrode was the cathode in sulfuric acid (3 M) and zinc concentration (100 g-Zn/L-3 M H_2_SO_4_) conditions. Activation and concentration polarization were measured by the potential sweep method. The measured values were converted to the log scale and then converted to a Tafel plot [16]. Electrode resistance was measured by the electrochemical impedance spectroscopy (EIS) method, and the modeling of the resistance circuit was conducted by the ZIVE ZMAN (EIS data analysis software) program.

## 3. Results and Discussion

### 3.1. Analysis of the Elecments Included in Electrode Composition

SEM and EDX analysis results of three-component (Ir + Sn + TaOx/Ti) and four-component (Ir + Sn + Ta + PdOx/Ti) insoluble catalyst electrodes are shown in Figure 1. The cross-section of the electrode coated with an insoluble catalyst material on the Ti substrate was measured by SEM and the catalyst component was analyzed using EDX, specifying a partial area of the coating material. Through the EDX qualitative analysis, it was confirmed that all of the elements included in the three-component and four-component systems were detected.

### 3.2. Overpotential and Electrode Resistance according to Electrode Composition 

Activation polarization and concentration polarization of the lead, Ir + Sn + TaOx/Ti (three-component), and Ir + Sn + Ta + PdOx/Ti (four-component) electrodes were calculated using the Tafel plot, as shown in Table 1. In the voltage–current relationship graph, the oxidation-reduction current interval appears. By logging the reduction current, a graph of the Tafel plot is obtained. When the applied current is low, it is greatly influenced by the electrode material; however, it is affected by the electrolyte concentration as the applied current is increased. As a result, the activation polarization according to the electrode material reactivity appears in the low applied current portion, while the concentration polarization according to the lower electrolyte concentration in the high applied current portion. Therefore, since activation and concentration polarization sections are divided according to the Tafel slope, the activation and concentration polarization values can be indirectly checked by checking the voltage of the sections (Figure 2) [16].

Activation polarization reflects an overpotential of charge transfer, which is more active when the activation polarization is low. The ion concentration is, therefore, greatly reduced locally, which in turn significantly increases the concentration polarization [16]. The activation polarization values of the lead, Ir + Sn + TaOx/Ti (three-component), and Ir + Sn + Ta + PdOx/Ti (four-component) electrodes were 0.09, 0.04, and 0.03 η_a_, respectively. IrOx improves the adhesive ability of coating materials on the Ti plate electrode, while PdOx is known to enhance the activation sites. The Ir + Sn + TaOx/Ti (three-component) and Ir + Sn + Ta + PdOx/Ti (four-component) electrodes showed a lower activation polarization by approximately 0.05 η_a_. The concentration polarization values of the lead, Ir + Sn + TaOx/Ti (three-component), and Ir + Sn + Ta + PdOx/Ti (four-component) electrodes were 0.40, 1.16, and 1.10 η_c_, respectively. The latter two electrodes with a lower activation polarization showed higher concentration polarization values than that of the lead (Pb) electrode. 

The overpotential of the electrode is sum of the activation polarization and concentration polarization,
(2)$$\eta =\eta_{a}+\eta_{c}.$$

According to the aforementioned results, the overpotential values of the lead, three-component, and four-component electrodes were calculated as 0.49, 1.20, and 1.13 η, respectively, indicating that the overpotential is greatly affected by the concentration polarization. A higher concentration polarization increased the rate of reduction of reactants [16], and thus zinc recovery was more active in the three- and four-component electrode systems compared to that in the lead electrode system.

Electrode resistance in lead was measured using the EIS method, as shown in Figure 3. The circuit diagram used in the resistance measurement is shown below, in which Rs is the solution resistance, Rct is the electrode resistance, W is the diffusion resistance, and Q is used in cases of a non-uniform electrode surface, instead of capacity (C).

Electrode resistance values, measured using the above resistance measurement circuit (Figure 3) of the lead, three-, and four-component electrodes were 5902.00, 0.98, and 0.02, respectively. The three- and four-component electrodes with IrOx and PdOx showed lower electrode resistance values than did the lead electrode (Table 1).

### 3.3. Comparative Evaluation of Recovery Yield according to Electrode Components

Metal zinc recovery using the lead, Ir + Sn + TaOx/Ti (three-component), and Ir + Sn + Ta + PdOx/Ti (four-component) electrodes was conducted by subjecting the reactants to a current so as to reduce zinc at the anode. Zinc recovery was conducted at various current densities (100, 200, 300, 400, and 500 A/m^2^) and the average voltage was identified for the given current density (Figure 4a). An increase in current density led to a greater increase in the average voltage in electrodes with a higher overpotential, and the slope values for the Pb, three-component, and four-component electrodes were 0.0006, 0.0014, and 0.0007, respectively. The changes in the slope for the three-component electrode increased rapidly after 200 A/m^2^, while the four-component electrode had a high overpotential of 1.13. Thus, the zinc recovery reaction occurs actively in the four-component electrode; however, its initial voltage is the lowest (2.48 V), while the change in the slope for the average voltage with the current density is not high. The four-component electrode is therefore the most suitable electrode to reduce the use of electrical energy.

The results of the zinc recovery yield according to the current density (Figure 4b) showed the highest yields in the lead and in the three- and four-component electrodes when the current density was 200 A/m^2^ and the recovery time was fixed at 4 h. The recovery yields were 99.41, 95.31, and 95.31%, respectively. As shown in the Galvanic measurement graph (Figure 5), when the initial voltage was low and the current density was 200 A/m^2^, after 3 h of recovery time the voltage increase was not significant in the four-component electrode (Figure 5c). Increasing the current density to over 200 A/m^2^ led to a significant increase in the initial voltage due to the significant increase in ion etching caused by the increase in current density, as well as a significant increase in the concentration polarization due to the reduction of the local ion concentration. The electrical energy graph associated with the current density also showed the lowest energy when the current density was 200 A/m^2^.

The theoretical amount of zinc recovery (g) can be calculated as below. This formula calculates the amount of recovered zinc by the electrochemical reaction according to the input current (Equation (3)):(3)$$\mathrm{Theoretical}\;\mathrm{amount}\;\mathrm{of}\;\mathrm{zinc}\;\mathrm{recovery}\;\left(g\right)=\frac{Zn^{2+}mol\left(mol\right)\times Zn\;atomic\;weight\left(g/mol\right)}{Zn^{2+}\;equivalent\;number},$$
(4)$$Zn2^{+}\;mol\left(mol\right)=\frac{Zn\;current\;\left(A\right)\times Time\;\left(h\right)}{Faraday\;constant\;\left(C/mol\right)}.$$

Electrical energy (kwh/ton-Zn) is the amount of energy required to recover zinc (Equation (5)):(5)$$Electrical\;energy\left(kwh/ton-Zn\right)=\frac{Zn\;current\;\left(A\right)\times Average\;voltage\;\left(v\right)\times Time\;\left(h\right)\times 1000}{Zn\;weight\;\left(ton\right)}.$$

On the other hand, lead electrodes are known to have high oxidation reactivity, and thus zinc recovery at the alumina (anode) is efficient. The lead electrode, however, had a high electrode resistance of over 5000 Ω compared to the three- and four-component electrodes, causing overpotential, which in turn leads to high electrical energy use. Consequently, at the same current density (200 A/m^2^), the amounts of electrical energy used by the lead, three-component, and four-component electrodes were 2498.97, 2391.40, and 2262.37 kwh/ton-Zn, respectively (Figure 4c). The use of the four-component electrode can reduce the electrical energy use by 10% compared to the amount required when using the lead electrode. Furthermore, insoluble electrodes cannot precipitate due to the presence of impurities in the electrolyte, further increasing the amount of electrical energy reduction when solvent extracts are used as electrolytes. Moreover, when zinc recovery using solvent extraction was conducted at the recovery condition (four-component electrode (cathode) and alumina (anode) as electrodes; current density: 500 A/m^2^; recovery time: 4 h), the electrical energy used by the four-component electrode was 2280.32 kwh/ton-Zn, corresponding to an approximately 19% decrease when compared to the lead electrode. Zinc recovery using a lead electrode leads to a reaction with impurities in electrolytes, which in turn results in precipitation to electrolytes, and such a electrochemical reaction with impurities has been observed to significantly reduce the yield. Although insoluble catalyst electrodes have an expensive disadvantage compared to Pb, the reason why they should be developed is that existing electrodes have a short lifespan of 1–2 years or less, while insoluble catalyst electrodes have a long lifespan of more than 5 years [14]. Moreover, in the actual electrolyte solution containing many impurities, the energy reduction was found to be 20% or more as compared with that in the Pb electrode.

### 3.4. Optimization of Zinc Recovery Condition according to Recovery Time and Current Density 

The activation, concentration polarization, electrode resistance, and recovery yield were measured by recovery time (1–4 h) of the four-component electrode, as shown in Table 2. The Figure 4c graph shows that the initial voltage of zinc recovery significantly increased at a current density of 200 A/m^2^, followed by a decrease in the voltage. Many factors increase the voltage, especially resistance and polarization. In Figure 6, zinc recovery yield is shown to be proportional to changes in the voltage. The initial recovery yield was low, about 79%, followed by an increase to about 95% after 3 h. This phenomenon was investigated through identifying the cause of the recovery yield increase by measuring activation, concentration polarization, and electrode resistance with the increase of the recovery time (Table 2).

During the initial 1 h of zinc recovery, the activation and concentration polarization significantly increased. This is attributed to the effect of crystallization polarization (η_crystal_), which occurs during the initial recovery of metal zinc at alumina (anode). The activation polarization significantly increased from 0.03 (initial) to 0.10 η_a_ during the first hour, followed by a decrease to 0.08–0.11 η_a_, and then a gradual increase. The increase in recovery time contributed to the increase of theactivation polarization from 0.03 to 0.11 η_a_, while the electrode resistance continuously increased from 0.02 to 0.25 Ω. With increasing zinc recovery, the activation and concentration polarization related to zinc recovery reactivity did not significantly increase; however, the electrode resistance increased due to the reduction in active surface area for the anode.

The recovery yield was therefore not significantly reduced according to recovery time, and the recovery yield increased to 95% after 3 h (Figure 6).

The results of the zinc recovery yield according to the current density are shown in Figure 4b. Zinc recovery was the highest at 200 A/m^2^, and the yield decreased with a further increase in the current density. To identify the phenomenon responsible for this result, the electrode resistance, activation, and concertation polarization were measured in the four-component electrode while increasing the current density for the initial hour. The results show that the highest recovery yield was obtained at a current density of 300 A/m^2^, and that the recovery yield decreased with the increase of the current density. A current density of 300 A/m^2^ (initial voltage: 2.62 V) increased the current flow, leading to a higher current density recovery rate compared to that obtained at 200 A/m^2^ (initial voltage: 2.56 V).

The scenario with increased recovery yield had a lower overpotential and smaller increase in activation polarization and electrode resistance, and the zinc recovery on the electrode surface was found to be optimum. There was thus hardly any increase in the initial overpotential during 3 h of zinc recovery in the optimum condition. Although a high current density of 300 A/m^2^ with a higher exerted current is effective, the increase in the electrode resistance decreased after 3 h of recovery time. The increase in the zinc recovery yield was higher at a current density of 200 A/m^2^ with less change in overpotential (Figure 7a and Table 3).

Furthermore, in order to confirm the optimal conditions for zinc recovery according to the current density, the recovery time was fixed at 1 h and zinc recovery was performed using the current density as a variable. Using this electrolytic solution, the electrode resistance was measured during the oxidation of the insoluble catalyst electrode, including the effect of the substance diffusion (Figure 7a). As a result, it was confirmed that the electrode resistance was greatly increased due to the decrease of the electrode reaction area, as hydrogen gas was generated at the electrode surface when the current density was 100 A/m^2^. Thereafter, the electrode resistance was greatly reduced when the current density was 200 A/m^2^. In addition, the electrode resistance was continuously increased due to the decrease of the substance diffusion, which resulted from the decrease of the zinc concentration as the current density was increased to 300–500 A/m^2^.

## 4. Conclusions


This study used an alumina (Al) anode, a lead electrode, and electrodes using an insoluble catalyst, which can increase the electrode stability and conductivity, as the cathode in order to increase the zinc recovery yield.The overpotential of the lead, three-component, and four-component electrodes were 0.49, 1.20, and 1.13 η, respectively, while the concentration polarizations were 0.40, 1.16, and 1.10 η_c,_. The overpotential was highly affected by the concentration polarization. A higher concentration polarization was found to generate a higher reduction in reactants, and thus the three- and four-component electrodes more actively recovered zinc compared to the lead electrode. Furthermore, the order of electrode resistance was the lead, the three-component, and the four-component electrodes.Using this sequence, the four-component electrode was identified as that with the greatest potential to lower the amount of required electrical energy due to its high reactivity for zinc recovery and low electrode resistance.At up to 3 h of zinc recovery time, the recovery yield was about 77% at the current density of 200 A/m^2^, which was lower than that at a current density of 300 A/m^2^ (recovery yield: about 80%). After 3 h of recovery time, however, the electrode resistance (reaction surface resistance due to zinc recovery) and the increase in overpotential with time were lower at a current density of 200 A/m^2^, significantly enhancing the zinc recovery yield to over 95%.In conclusion, an increase in recovery yield is accompanied by a lower overpotential. As zinc recovery occurs on the electrode surface, the electrode with a smaller increase in activation polarization and resistance was determined to be more effective for zinc recovery.


## Figures and Tables

**Figure 1 materials-12-02779-f001:**
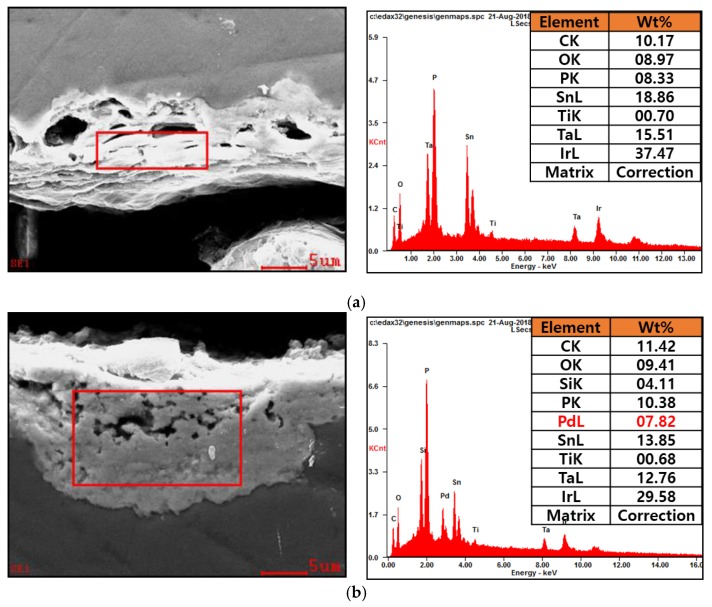
SEM and EDX analysis of Ir + Sn + TaOx/Ti (**a**) and Ir + Sn + Ta + PdOx/Ti (**b**).

**Figure 2 materials-12-02779-f002:**
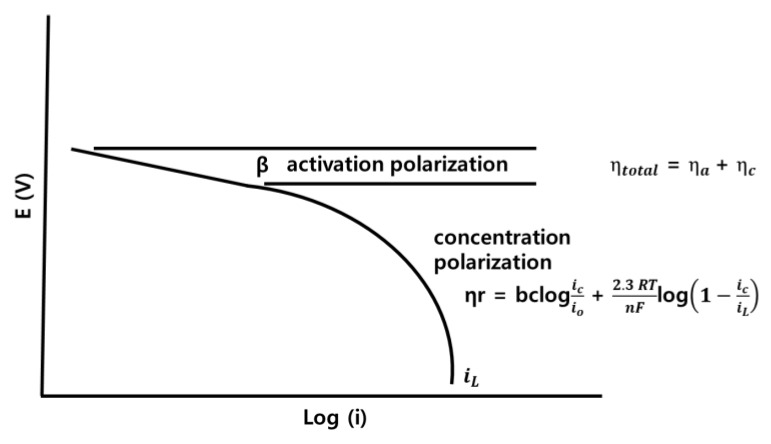
Tafel plot of activation and concentration polarization.

**Figure 3 materials-12-02779-f003:**
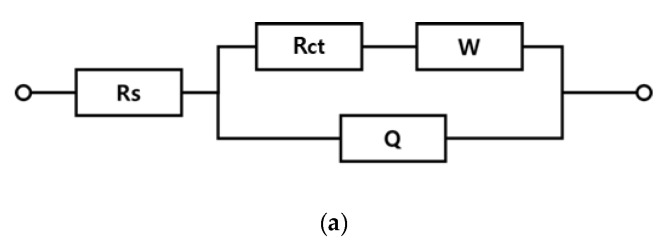
Circuit diagram used in resistance measurement (**a**), Nyquist plot of Pb (**b**), and that Ir + Sn + TaOx/Ti and Ir + Sn + Ta + PdOx/Ti electrodes (**c**).

**Figure 4 materials-12-02779-f004:**
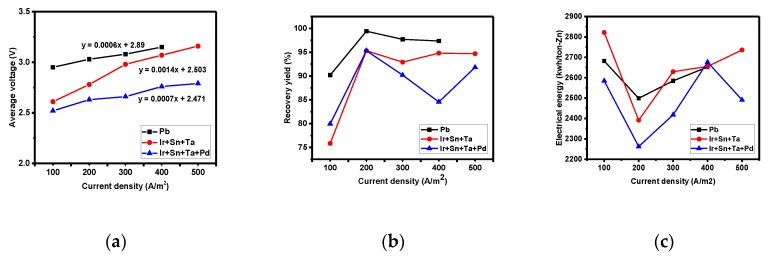
Average voltage (**a**), recovery yield (**b**), and electrical energy (**c**) by current density.

**Figure 5 materials-12-02779-f005:**
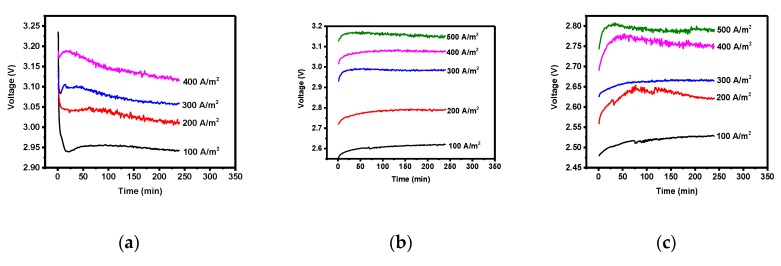
Galvanic graph for Pd (**a**), Ir + Sn + TaOx/Ti (**b**), and Ir + Sn + Ta + PdOx/Ti (**c**).

**Figure 6 materials-12-02779-f006:**
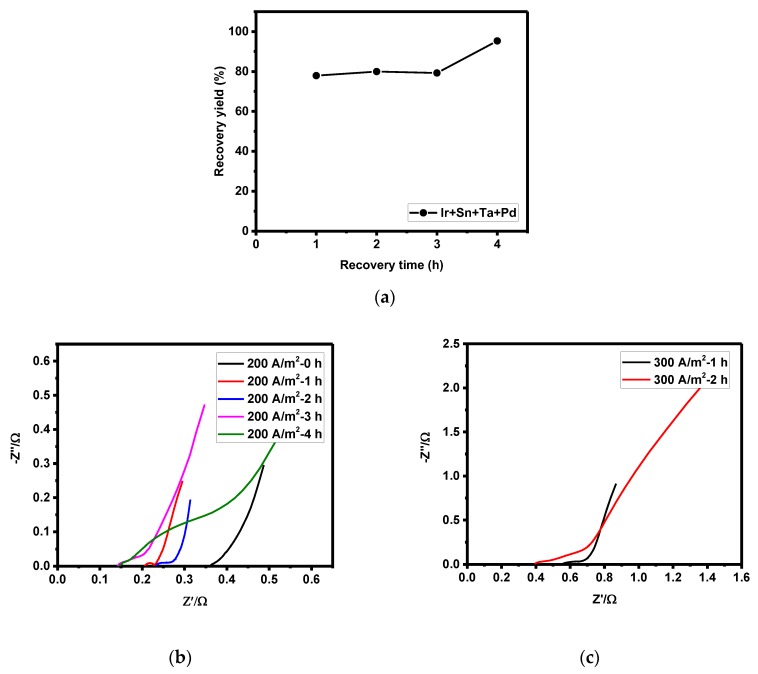
Recovery yield in the four-component electrode according to recovery time (**a**), Nyquist plot of Ir + Sn + Ta + PdOx/Ti at a current density of 200 A/m^2^ according to recovery time (0–4 h) (**b**), and at a current density of 300 A/m^2^ according to recovery time (1–2 h) (**c**).

**Figure 7 materials-12-02779-f007:**
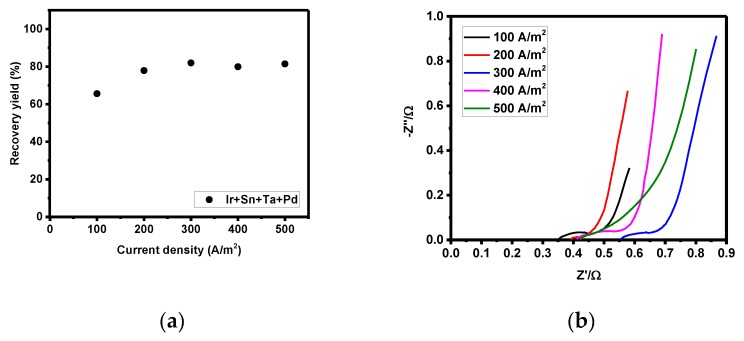
Recovery yield in the four-component electrode according to current density (**a**), and Nyquist plot of Ir + Sn + Ta + PdOx/Ti according to current density (0–500 A/m^2^) (**b**).

**Table 1 materials-12-02779-t001:** Overpotential and electrode resistance by electrode components.

Sample	Solution Condition	Activation Polarization (η_a_)	Concentration Polarization (η_c_)	Activation + Concentration Polarization (η)	Electrode Resistance (Rct, Ω)	Solution Conductivity at 25 °C (mS/cm)
**Pb**	**Sulfuric acid-3 M, Zn-100 g/L**	0.09	0.40	0.49	5902.00	405.20
**Ir + Sn + Ta**		0.04	1.16	1.20	0.98	
**Ir + Sn + Ta + Pd**		0.03	1.10	1.13	0.02	

**Table 2 materials-12-02779-t002:** Overpotential and electrode resistance in the four-component electrode according to recovery time.

Sample	Current Density (A/m^2^)	Recovery Time (h)	Activation Polarization (η_a_)	Concentration Polarization (η_c_)	Activation + Concentration Polarization (η)	Electrode Resistance (Rct, Ω)	Solution Conductivity at 25 °C (mS/cm)
**Ir + Sn + Ta + Pd**	200	0	0.03	1.10	1.13	0.02	446.30
		1	0.10	1.30	1.40	0.03	422.70
		2	0.08	1.29	1.37	0.06	394.10
		3	0.10	1.30	1.40	0.24	411.70
		4	0.11	1.29	1.40	0.25	409.20
	300	1	0.11	1.28	1.39	0.11	416.40
		2	0.10	1.29	1.39	0.13	373.30

**Table 3 materials-12-02779-t003:** Overpotential and electrode resistance in the four-component electrode according to current density.

Sample	Current Density (A/m^2^)	Activation Polarization (η_a_)	Concentration Polarization (η_c_)	Activation + Concentration Polarization (η)	Electrode Resistance (Rct, Ω)	Solution Conductivity at 25 °C (mS/cm)
**Ir + Sn + Ta + Pd**	0	0.03	1.10	1.13	0.02	446.30
	100	0.04	1.29	1.33	0.10	436.50
	200	0.10	1.30	1.40	0.03	422.70
	300	0.11	1.28	1.39	0.11	416.40
	400	0.06	1.30	1.36	0.13	415.20
	500	0.12	1.28	1.40	0.27	402.70

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
