# Peer review of "Zinc Recovery through Electrolytic Refinement Using Insoluble Ir + Sn + Ta + PdOx/Ti Cathode to Reduce Electrical Energy Use"

_materials, 2019, doi:10.3390/ma12172779_

Round 1
Reviewer 1 Report
Expand on the introduction. It looks like a lab report. Do not reflect thorough research.
Not much detailed explanation provided by the authors in the electrochemical characterization.
Show the Nyquist plot with circuit fit. The R is the polarization resistance or charge-transfer resistance at the electrode/solution interface according to Randle circuit. It is not electrode material resistance. Charge transfer resistance (Rct) is the reaction resistance at the limit of infinite frequency and zero amplitude AC perturbation. Polarization resistance (Rp) is the reaction resistance at the limit of ZERO frequency and zero amplitude perturbation; it is at low-frequency end and small amplitude perturbation. Polarization resistance signifies the barrier between the cathodic and anodic current flow throughout the electrode configuration (2 or 3 electrode system). Mainly the charge-transfer resistance (Rct) and diffusion resistance (Rw). Rp=Rct+Rw. Change the R to Rct.
Show in a figure (main text) how activation and concentration polarization calculated from Tafel slope in your case. Explain it for readers in the writing.
No other material characterization is provided (XRD, SEM, and XPS).
Author Response
Q1> Expand on the introduction. It looks like a lab report. Do not reflect thorough research.
Answer>
Thank you for your comments, we answered it and it was added in section 1. Introduction.
Design factors for the zinc recovery process include zinc purity, cathode cell voltage, electrical energy, pH, current density, zinc concentration, recovery temperature, and electrode spacing. The factor for pH, zinc solution was prepared as using zinc oxide and sulfuric acid, and zinc concentration was reported to be different depending on the pH. The reaction scheme of the reaction is as follows (equation 1).
ZnO + H2SO4 → ZnSO4 + H2O (1)
Pearson et al. (1981) showed zinc leaching concentrations according to pH in industrial dust containing zinc by EAFD method, when the pH was 2, zinc was recovered to 85-90%, and the zinc recovery was reduced by 80% as the pH was increased to 3-4 [15]. By lowering the pH, a high concentration of zinc solution could be produced. In addition, current density, zinc concentration, recovery temperature, and electrode spacing, etc. were known to promote zinc recovery by improving Zn ion transfer [15].
Q2> Not much detailed explanation provided by the authors in the electrochemical characterization.
Show the Nyquist plot with circuit fit. The R is the polarization resistance or charge-transfer resistance at the electrode/solution interface according to Randle circuit. It is not electrode material resistance. Charge transfer resistance (Rct) is the reaction resistance at the limit of infinite frequency and zero amplitude AC perturbation. Polarization resistance (Rp) is the reaction resistance at the limit of ZERO frequency and zero amplitude perturbation; it is at low-frequency end and small amplitude perturbation. Polarization resistance signifies the barrier between the cathodic and anodic current flow throughout the electrode configuration (2 or 3 electrode system). Mainly the charge-transfer resistance (Rct) and diffusion resistance (Rw). Rp=Rct+Rw. Change the R to Rct.
Answer>
Thank you for your comments, we added Nyguist plots of the each samples measuring resistance, and modified R to Rct.
Q3> Show in a figure (main text) how activation and concentration polarization calculated from Tafel slope in your case. Explain it for readers in the writing.
Answer>
Thank you for your comments, we answered it and it was added in section 3.2.
In the voltage-current relationship graph, the oxidation-reduction current interval appears. By logging the reduction current, a graph of the Tafel plot is obtained. When the applied current is low, it is greatly influenced by the electrode material, however it is affected by the electrolyte concentration as the applied current is increased. As a result, the activation polarization according to the electrode material reactivity appears in the low applied current portion, the concentration polarization according to the lower electrolyte concentration in the high applied current portion. Therefore, since activation and concentration polarization sections are divided according to the Tafel slope, the activation and concentration polarization values ​​can be indirectly checked by checking the voltage of the sections [16].
Q4> No other material characterization is provided (XRD, SEM, and XPS).
Answer>
Thank you for your comments, we answered it and it was added in section 3.1.
SEM and EDX analysis results of three-component (Ir+Sn+TaOx/Ti) and four-component (Ir+Sn+Ta+PdOx/Ti) insoluble catalyst electrodes were shown as follows (Figure 1). The cross section of the electrode coated with an insoluble catalyst material on the Ti substrate was measured by SEM, the catalyst component was analyzed using EDX specifying a partial area of ​​the coating material. Through the EDX qualitative analysis, it was confirmed that all of the elements included in the three-component and four-component systems were detected.
Reviewer 2 Report
According to the paper, the aim of the research is to reduce the cost of the Zn recovery production by changing the Pd electrodes. Indeed, the authors achieved a 10% energy reduction, but they used electrodes with Ir, Sn, Ta, PdO, Ti, which are rather expensive materials themselves.
To my opinion the paper is not particularly interesting to the general reader of the "Materials" journal and should be published elsewhere.
Author Response
Q1> According to the paper, the aim of the research is to reduce the cost of the Zn recovery production by changing the Pd electrodes. Indeed, the authors achieved a 10% energy reduction, but they used electrodes with Ir, Sn, Ta, PdO, Ti, which are rather expensive materials themselves.
Answer>
Thank you for your comments, we answered it and it was added in section 3.3.
Although insoluble catalyst electrodes have an expensive disadvantage compared to Pb, the reason why it should be developed is that existing electrodes have short lifespan of 1-2 years or less, while insoluble catalyst electrodes have long lifespan of more than 5 years [14], in the actual electrolyte solution containing a lot of impurities, the energy reduction was found to be 20% or more as compared with the Pb electrode.
Round 2
Reviewer 1 Report
good job.
Reviewer 2 Report
Thank you for the revision